# An Intersection Signal Control Mechanism Assisted by Vehicular Ad Hoc Networks

**Zhen Cai [1,2,]\*, Zizhen Deng [1], Jinglei Li [1], Jinghan Zhang [1] and Mangui Liang [2]**

1    Department of Computer, North China Electric Power University, Baoding 071003, China;
     dengzizhen@ncepu.cn (Z.D.); lijinglei@ncepu.edu.cn (J.L.); jhzhang@ncepu.edu.cn (J.Z.)
2    Institute of Information Science, Beijing Jiaotong University, Beijing 100044, China; mgliang@m.bjtu.edu.cn
\*    Correspondence: zhencai@ncepu.edu.cn

**Abstract:** The urban intersection signal decision-making in traditional control methods are mostly based on the vehicle information within an intersection area. The far vehicles that have not reached the intersection area are not taken into account, which results in incomplete information and even incorrectness in decision-making. This paper presents an intersection signal control mechanism assisted by far vehicle information. Using the aid of real-time information collection for far vehicles through vehicular ad hoc networks (VANETs), we can consider them together and calculate the accumulative waiting time for each intersection traffic flow at a future moment to make the optimal signal decision. Simulation results show that, under three different traffic flow environments—same even traffic flows, same uneven traffic flows, and different traffic flows—the two proposed implementation schemes based on the mechanism (fixed phase and period timing improvement scheme, and dynamic phase and period control scheme) show good performances, in which the average waiting time and the ratio of long-waiting vehicles are both less than the results of the traditional signal timing scheme. Especially, in the second scheme, the waiting time was reduced by an average of 38.6% and the ratio of long-waiting vehicles was reduced by an average of 7.67%.

**Keywords:** Intelligent transportation; signal control; waiting time; far vehicle; VANETs

## 1. Introduction

An efficient traffic management system is the base and essential prerequisite for economic and social development. Intersection management is one of the most challenging issues in urban transportation. The area ratio of intersections in urban road networks is too small; however, according to statistics, 64.31% of urban road traffic accidents in China occur at intersections and their vicinity [1], and the time lost in vehicle trips at intersections accounts for the largest part of the total time lost. A sophisticated and reliable intersection management mechanism can greatly increase the urban transportation efficiency and ensure people's travel safety.

The research on intersection signal control has been developing and evolving in recent years. In terms of the control range, it has gradually evolved from a single intersection to multiple intersections and an urban region [2,3]. In terms of control methods, it has developed from the traditional fixed period timing to the full induction and adaptive control methods [4–6]. Li et al. [7] proposed a signal control optimization model for urban multi-intersections using three-phase theory and information-exchanging. Wang et al. [8] proposed a group-based signal optimization model that considers both the safety and delay for the intersections with mixed traffic flows. García-Nieto et al. [9] designed an optimization approach in which a particle swarm optimizer (PSO) can find successful traffic light cycle programs. A multi-stage decision model to minimize the waiting time is proposed in Wang et al. [10]. The model takes advantages of the structure of the minimum green and red time, which significantly reduces

the model scale by properly choosing the system states and control actions. In order to alleviate its traffic pressure effectively, a coordinated arterial traffic type-2 fuzzy logic control (FLC) method [11] is proposed. The method allocates green time according to the traffic situation of each intersection; meanwhile it adjusts each intersection's green time based on the vehicles between the intersection and the downstream intersections for the purpose of enlarging the green wave band. Lu et al. [12] proposed an optimization model for network progression coordinated control that enables the phase time and phase sequence of each intersection to be freely valued within the solution space, and realizes the comprehensive optimization of the phase time, as well as a common signal cycle, phase sequence, and offset. Sanchez-Iborra et al. [13] proposed an adaption of RED (random early detection) by incorporating an active queue management technique inherited from data networks into vehicle traffic management at the signalized intersections.

A vehicular ad hoc network [14] (VANET) is based on the concept of an Internet of Vehicles (IoV) [15] in the category of network forming. As a special type of mobile ad hoc network (MANET), VANET has unique characteristics, such as a high mobility of nodes, restricted forwarding directions, and many influencing factors. The vehicle to vehicle (V2V) communication and the communication between vehicles and the infrastructure (V2I) provide guarantees for the accurate, real-time, and comprehensive information acquisition and dissemination in an intelligent transportation system (ITS) [16–20]. It plays an active role in improving the quality of urban transportation services. Al Mallah et al. [21] proposed a framework for the real-time distributed classification of congestion into its components on a heterogeneous urban road network using a VANET. Lin et al. [22] synthesized and studied methods that are used to optimally group autonomously controlled vehicles so they can travel along a highway in platoons. Vehicular formations are structured to yield an effective autonomous mobility operation and to realize high-performance multihop dissemination of multiclass messaging flows. Oche et al. [23] presented a comprehensive survey of Quality of Service (QoS)-aware routing protocols and examined the protocols based on their ability to support ITS infotainment services; their multi-constraint path problem (MCP); and the protocol's functionality and weaknesses, objectives, and design challenges.

VANETs have been widely applied in the field of intersection signal control as well. Chanana et al. [24] realized a form of traffic navigation communication through the aid of V2V, which reduced the risk of intersection collisions and significantly improved the throughput of intersections. Sharma et al. [25] improved the inter-vehicle communication capability by using the Ad hoc On-demand Distance Vector routing protocol AODV in a VANET at the intersection, based on which, the signal control system can effectively alleviate the congestion of an intersection. In Lin et al. [26], a traffic flow zoning control idea and a mathematical model of the system optimization at intersections is proposed based on the analysis of the environmental technical characteristics of the IoV. Lin et al. [27] proposed a novel coordination method for intersection management in a connected vehicle environment in which the road network is divided into three logical sections, i.e., a buffer area, core area, and free driving area. Chang et al. [28] proposed a traffic signal control algorithm that assigns intersection vehicles to the group of each lane and calculates the traffic volume and congestion degree using the traffic information of each group through inter-vehicle communication in a VANET. Hemakumar et al. [29] proposed an adaptive and intelligent traffic control system based on a VANET in which a traffic signal controller is placed at each intersection. The system uses it to communicate with the vehicles in the area, and calculate the priority of each signal phase according to the waiting time.

In this study, aiming at the issue of optimal decision-making for intersection signal control [11,30–32], we designed a real-time information collection scheme for the far vehicles using a VANET, and took them into consideration, as well as the information of the intersection vehicles, to determine the intersection traffic condition at a certain time in the future. According to the calculated waiting time of each traffic flow, the signal decision could be made more precisely. In addition, two implementation schemes based on the signal control mechanism were proposed, i.e., fixed phase and period timing improvement scheme, and a dynamic phase and period control scheme. Simulation results showed that

these schemes effectively reduced the average waiting time of vehicles and the ratio of long-waiting vehicles, and outperformed the traditional signal control method significantly.

The rest of this paper is organized as follows. In Section 2, we describe how to collect the information of far vehicles using a VANET. In Section 3, we present the calculation method for the waiting time of traffic flows and propose the two concrete decision-making schemes. The simulation results and performance analysis are presented in Section 4. The last section concludes this paper. In order to enhance the readability of this paper, the summary of presented symbols is shown in Table 1.

**Table 1.** Summary of presented symbols.

| Notation | Description |
| --- | --- |
| $t_c$ | Current moment |
| $t_f$ | Future moment |
| $n_{max}$ | Capacity of the road |
| $n_{id}$ | Number of vehicles in the "island" |
| $v^{t_c}$ | Real-time velocities at $t_c$ |
| $pos^{t_c}$ | Real-time positions at $t_c$ |
| $T_{ls}$ | The time that the vehicle has been waiting for |
| $T_d^j$ | Estimated waiting time of vehicle $j$ |
| $WT_i$ | Estimated waiting time of traffic flow $i$ |
| $P_i$ | $i$th signal phase |
| $qn_i^{t_f}$ | Number of vehicles in flow $i$ that have stopped at $t_f$ |
| $k_i^{T_g}$ | The vehicles in the $i$th traffic flow that can pass the intersection in $T_g$ |
| $T_d'$ | Adjusted waiting time |
| $ET_{P_i}^{t_f - t_e}$ | The eliminable waiting time of the phase $P_i$ |

## 2. Information Collection for Far Vehicles

So far, in most of the scientific research and engineering applications, the traffic signal control system at the urban intersection generally collects the information of vehicles within the intersection area as the input data for modeling and calculation. The data can be easily acquired and then preliminarily analyzed using visual sensor devices at the intersection (e.g., cameras) and some corresponding image processing techniques. However, vehicles that have not entered the intersection area before the decision-making time may reach the intersection later, potentially impacting the effectiveness and accuracy of the decision. As shown in Figure 1, at time $t_1$, based on the information of vehicles within the intersection area, the signal control system makes the decision to change to a green light for the north-south bidirectional traffic flows. At a later time $t_2$, a large number of vehicles on the east side enter the intersection area and encounter the forbidden signal. They have to wait for the right-of-way, which results in an increase in overall delay time and a decline in traffic capacity at the intersection. Therefore, it is clear that the information of the vehicles in the non-intersection area (which we call far vehicles in this paper) is of great significance for the signal control decision.

In this paper, a VANET was used to collect the real-time information of the far vehicles on the road. The assumptions and method were as follows.

It was assumed that all vehicles are equipped with the on-board equipment supporting V2V and V2I communication. The roadside unit (RSU) that can communicate with vehicles is deployed at each intersection to collect and process the real-time information of vehicles, and then provide them to the signal control system for decision-making; each RSU can directly communicate and exchange data with its adjacent RSUs through wired or cellular networks.

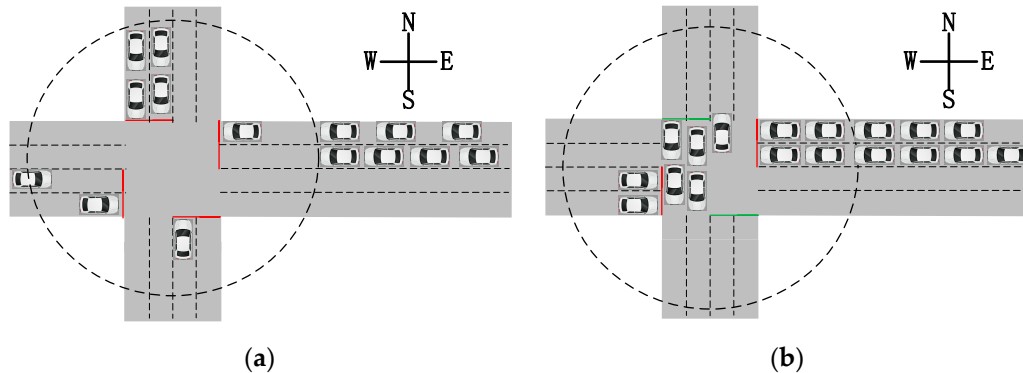

**Figure 1.** Impact of far vehicles on intersection signal decision-making: (**a**) intersection situation at $t_1$ and (**b**) intersection situation at $t_2$.

For vehicle information collection, we assumed that each vehicle is equipped with the Global Position System (GPS) and the digital map, and then it can easily acquire the information about its own position, velocity, moving direction, etc. With the aid of the periodical beacon messages (which are mandatory in most of VANETs protocols, e.g., 802.11p), which are exchanged with each other, and the information can also be obtained by their neighboring vehicles within the transmission range. Each RSU sends a dedicated packet, i.e., an ICP (information collection packet), to each adjacent intersection through a VANET at fixed time intervals. The ICP is relayed using a multi-hop connection through all vehicle nodes on the road sequentially. After any vehicle-node receives it, it attaches its information including its own ID, current velocity ($v$), current position ($pos$), acceleration ($acc$), deceleration ($dec$), desired velocity ($v_d$), and vehicle length ($clen$) to the packet. Then, it selects the closest vehicle in the direction of the packet forwarding as the next hop node to relay the ICP to. If the multi-hop connection through the whole road is active, which means the distance between any two successive vehicles is less than the transmission range of the VANET ($R$), the ICP can reach the RSU at the destination intersection smoothly. Then, the information of all vehicles on the road is packed and sent back to the source intersection RSU via their own networks (Figure 2a). Each intersection maintains a vehicle information table that records and updates the information of each running vehicle in real-time, and adds a time stamp ($t_s$) to record the time point of information gathering.

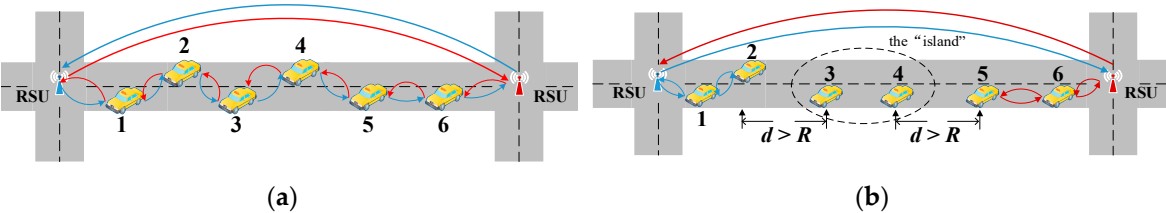

**Figure 2.** Collection of real-time vehicle information through a vehicular ad hoc network (VANET): (**a**) end-to-end connecting using multi-hops and (**b**) breaking by "island." RSU: roadside unit.

When there is a small amount or an uneven distribution of vehicles on the road, the distance $d$ between some pairs of successive vehicles may be greater than the transmission range $R$, which means the ICP cannot continue to be forwarded right along the road and a network "island" occurs. In this case, the vehicle node that holds the ICP relays it back immediately to the source RSU via the original path. As shown in Figure 2b, the vehicles 3 and 4 are in the range of the "island" and cannot communicate with the two side RSUs. When the ICPs from the RSUs on both ends arrive at vehicle 2 and vehicle 5, respectively, and cannot be forwarded continually due to the distance between them being longer than $R$, they are sent back at once. In order to solve this problem of incomplete information collection for all vehicles on the whole road, we need to estimate the states of vehicles in

the "island" range based on their available existing data. We take the condition in Figure 2b as an instance to describe the estimation process. First, the range of the "island" needs to be determined. We assumed that the center position of the left intersection is the zero point and the value of the position increases toward the right. Then, the value of the center position of the right intersection is *slen*, which is considered equal to the road length, and the "island" range is ($pos_2 + R$, $pos_5 - R$), where $pos_2$ and $pos_5$ are the positions of vehicles 2 and 5, respectively, in the figure. Second, we estimated the velocity of the vehicle $v_e$ in the "island" by considering the traffic density on the road [33] using Equation (1):

$$v_e = v_d \Big/ \left\{ 1 + a \times \left[ \frac{n_{id} \times \left( \frac{slen}{pos_5 - pos_2} \right)}{n_{max}} \right]^b \right\} \tag{1}$$

where $n_{max}$ is the capacity of the road, $n_{id}$ is the number of vehicles in the "island," and $a$ and $b$ are adjustable parameters. Then, according to the "losing duration" $t_s - t_c$ and the current position ($pos'$) can be calculated, where $t_s$ is the timestamp of its latest record and $t_c$ is the current time. Moreover, as the vehicle movement is affected by the interactions with the vehicles around it and many other factors, which are hardly recognized quantitatively, such as the road condition, weather, driver's mental status, and even whether it is a light traffic flow. In some cases, the calculated positions of some vehicles are out of the "island" range (i.e., $< pos_2 + R$ or $> pos_5 - R$). In other words, the estimated result is contrary to the actual situation. In this case, in order to make the result more accurate and realistic, we need to correct them using Equation (2), where we set the out-of-range estimated position as its nearest boundary value of the "island," i.e., they are pulled back into the "island" where they really are:

$$pos^{t_c} = \begin{cases} pos_2 + R & v_e \times (t_s - t_c) < pos_2 + R \\ v_e \times (t_s - t_c) & pos_2 + R \le v_e \times (t_s - t_c) \le pos_5 - R \\ pos_5 - R & v_e \times (t_s - t_c) > pos_5 - R \end{cases} \tag{2}$$

Finally, after the calculation, the RSU updates the current position ($pos^{t_c}$) of each "island" vehicle in the vehicle information table. At this point, the RSUs on both ends can obtain the information of all vehicles on the road in real-time.

## 3. Intersection Signal Control Mechanism

### 3.1. System Model and Environment Settings

Figure 3 illustrates a typical urban intersection. Without the right turning, which generally has its exclusive lanes out of control of the signal lights at most current urban intersections, the number of traffic flows ($m$) is eight. $D_i(i \in [1, m])$ represents the $i$th flow, and the value of $p_{ij}$ indicates whether the $i$th and the $j$th traffic flows can pass the intersection at the same time ($p_{ij} = 1$ ($i \ne j$) means no conflict, and vice versa). In a traditional signal control mechanism, according to the eight traffic flows as shown in Figure 3a, there are four signal phases, consisting of north-south bi-directional straight paths, east-west bi-directional straight paths, north-east and south-west left turnings, and west-north and east-south left turnings. However, in the condition of a dynamic combination with no conflict, 12 phases can be realized in Figure 3b:

$$\sum_{i,j \in [1, m]} p_{ij} = 12, \ (i \ne j). \tag{3}$$

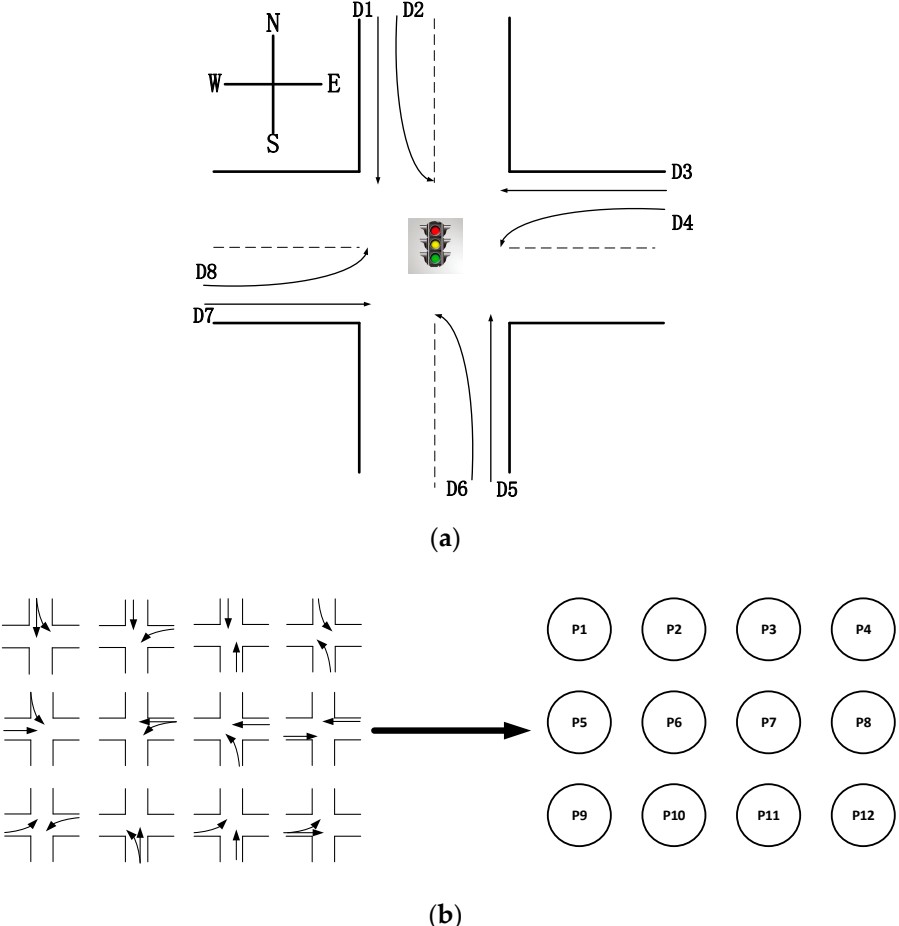

**Figure 3.** Abstraction of signal phases at the intersection: (**a**) urban intersection diagram and (**b**) the 12 combinations of traffic flows.

The green light duration ($T_g$) can be determined dynamically within the range of $[\underline{T_g}, \overline{T_g}]$ via the signal control system, and the yellow light duration ($T_y$) is set as a fixed value. Due to the different width and different number of lanes, and other conditions of each traffic flow, the maximum number of vehicles ($\overline{k_i}$) that can pass in a unit time is different and the actual value can be acquired according to the statistical data of the intersection. The moving direction (i.e., driving path) of far vehicles after they reach the intersection can be gathered in advance by also using a VANET. However, in consideration of the personal privacy, instead, far vehicles can be divided according to the proportion of each flow in the recently collected statistical data.

In addition, for the state of the traffic flow at a future moment, the calculation in our proposed mechanism adopts the method of forward recurrence based on the information at the current time. If the traffic volume is large, the amount of calculations required is great as well. Considering that the hardware performance of each decision-making device at different intersections is not same, we can appropriately adjust the decision time point ($t_e$, $t_e \geq t_c$) forward from the end time of the current signal according to the actual condition. In other words, the decision-making needs to be started in advance to ensure that the calculation is completed on time.

### 3.2. Waiting Time at a Future Moment

In the decision-making process, the control system first assumes that signals for all traffic flows are red lights. Then, based on the real-time information of far vehicles and intersection vehicles (i.e., vehicles within the intersection area), it can calculate the waiting time ($T_s$) at a future moment

($t_f$) for each vehicle. Note that in this paper, the waiting time of vehicle means the duration from the moment of vehicle's first stop due to the red light or the front waiting queue to $t_f$, i.e., how long the vehicle will have waited for. Furthermore, the waiting time of each traffic flow ($WT_i$) can be calculated cumulatively.

For vehicle $j$ that has stopped in the intersection area before the current time ($t_c$), it is easy to compute its waiting time ($T_d^j$) at a future moment $t_f$ using Equation (4):

$$T_d^j = T_{ls} + \left(t_f - t_c\right), \tag{4}$$

where $T_{ls}$ is the length of the time that the vehicle has been waiting for up to the present moment. Furthermore, using visual information gathering, the average single-lane queue length ($len_i$) of each traffic flow can be easily obtained as well.

For unstopped intersection vehicles (i.e., their velocities are greater than 0) and far vehicles, we need to further estimate their positions and states at $t_f$ (i.e., whether they have stopped and entered the waiting queues) according to existing collected information. First, we use their real-time velocities ($v^{t_c}$) and positions ($pos^{t_c}$) at $t_c$ to calculate the positions of all vehicles for the next unit time ($pos^{t_c+1}$) according to:

$$pos^{t_c+1} = pos^{t_c} + v^{t_c} \times \Delta t, \tag{5}$$

where $\Delta t$ is the unit time for calculating. Furthermore, considering that the vehicle wants to stop at the tail of the front queue, a deceleration process is required in advance. If at the moment of $t_c + 1$, $pos^{t_c+1}$ has entered the general deceleration range, the value of its velocity at the next unit time ($v^{t_c+1}$) is adjusted according to its deceleration ($dec$):

$$v^{t_c+1} = \begin{cases} v^{t_c} & pos^{t_c} + \left(\frac{v^{t_c 2}}{2dec}\right) \le qlen \\ v^{t_c} - dec & pos^{t_c} + \left(\frac{v^{t_c 2}}{2dec}\right) > qlen \end{cases}. \tag{6}$$

In addition, if $pos^{t_c+1}$ falls within the range of the front queue in the traffic flow that the vehicle belongs to (i.e., $pos^{t_c+1} > qlen_i$), it is determined that it has stopped and entered the queue. Then, that time is considered its stop time ($t_s^j$), and the waiting time at $t_f$ of vehicle $j$ can be calculated using:

$$T_d^j = t_f - t_s^j. \tag{7}$$

Besides that, we need to update the average single-lane queue length of the corresponding traffic flow ($qlen_i$). The vehicle length ($clen^j$) and the safe stopping distance ($sd$) should be considered. The new average length of the flow can then be calculated as follows:

$$qlen_i = qlen_i + \left(clen^j + sd\right)/n_i, \tag{8}$$

where $n_i$ is the number of lanes that belong to the $i$th traffic flow at the intersection. If $pos^{t_c+1}$ does not fall within the range of the front queue, i.e., $pos^{t_c+1} \le qlen_i$, we need to continue to calculate its position at $t_c + 2$ (i.e., $pos^{t_c+2}$) based on the last value, update its velocity according to Equation (6), and judge whether it has stopped.

Finally, the sum of the cumulative waiting time of all stopped vehicles is calculated for each traffic flow until the time $t_f$ using:

$$WT_i = \sum_{j=1}^{qn_i^{t_f}} T_d^j, \tag{9}$$

where $qn_i^{t_f}$ is the number of vehicles in flow $i$ that have stopped at $t_f$.

### 3.3. Fixed Phase and Period Signal Timing Improvement Scheme

The traditional signal timing model adopts the decision mechanism of dynamically allotting an appropriate time within the fixed period to the signal phase of each traffic flow according to the conditions of the vehicles in the intersection area in the present. By incorporating the information of far vehicles, in this section, the fixed phase and period signal timing improvement scheme is proposed based on the waiting time of each traffic flow that we have discussed above.

For the intersection in Figure 3, based on the common traditional classification, there are four phases, i.e., P1 (D1&D5), P2 (D2&D6), P3 (D3&D7), and P4 (D4&D8). As such, we can calculate the waiting time of each phase using Equation (10):

$$\begin{cases} WT_{P1} = WT_1 + WT_5 \\ WT_{P2} = WT_2 + WT_6 \\ WT_{P3} = WT_3 + WT_7 \\ WT_{P4} = WT_4 + WT_8 \end{cases}, \tag{10}$$

and the total waiting time for all phases can be calculated using:

$$WT_s = \sum_{i=1}^{8} WT_i. \tag{11}$$

Then, the green light time $T_{g-p_i}$ can be allotted to each signal phase according to the proportion of the waiting time of each phase using:

$$T_{g-p_i} = \frac{WT_{P_i}}{WT_s} \times \left( T_p - T_y \times 4 \right), \tag{12}$$

where $T_p$ is the length of a complete signal period and $T_y$ is the length of a yellow light period.

### 3.4. Dynamic Phase and Period Signal Control Scheme

In contrast to the fixed phase and period signal timing scheme, in this section, we propose another signal control scheme that can dynamically make the optimal decision regarding the combination phase and the green light duration. The decision-making system adopts the sum of the waiting time of vehicles that can pass the intersection during the green light time as the evaluation reference, which we call the eliminable waiting time. In addition, for an individual vehicle that has waited a long time for a green light, we set a weighting factor that can increase with the waiting time to ensure the driver passes the intersection.

#### 3.4.1. Eliminable Waiting Time of the Traffic Flow

Under the situation that there is a long waiting queue, not all waiting vehicles can pass the intersection within the next green light time (even with the maximum possible value). Therefore, sometimes it is not applicable and appropriate to use the total waiting time of all vehicles in the queue as the evaluation reference. Based on this point, the solution in the scheme is to provide an eliminable waiting time ($ET_i^{T_g}$). It is the sum of the waiting time of vehicles (in the $i$th traffic flow) that can pass the intersection during the given green light period $T_g$. In other words, we should instead consider how much waiting time can be eliminated in the next green light period.

First, we need to determine the maximum number of vehicles in the $i$th traffic flow ($k_i^{T_g}$) that can pass the intersection in $T_g$ in the current environment using:

$$k_i^{T_g} = T_g \times \overline{k_i}, \tag{13}$$

where $\overline{k_i}$ is the maximum number of vehicles that can pass in a unit time (as described in Section 3.1). Compared with the unstopped vehicles, a vehicle that has stopped in the waiting queue before the decision-making time ($t_c$) can pass the intersection earlier. If the number of stopped vehicles $q_i$ is greater than (or equal to) $k_i^{T_g}$, the front $k_i^{T_g}$ vehicles in the queue should be considered. The sum of their waiting time, which is the eliminable waiting time of traffic flow $i$, is calculated as follows:

$$ET_i^{T_g} = \sum_{j=1}^{k_i^{T_g}} T_d^j, \ q_i \geq k_i^{T_g}. \tag{14}$$

If $q_i < k_i^{T_g}$, this means that all the vehicles in the queue can pass the intersection in $T_g$. In this case, we should take the unstopped vehicles that can reach and pass the intersection into account. Based on the vehicle information collected using the VANET, we can calculate the position ($pos_{t_f}$) of the vehicle at $t_f$ using Equation (15) if there are no waiting vehicles ahead, and as such, it does not have to stop:

$$pos_{t_f} = \begin{cases} \left(t_f - t_c - \left(R - pos^{t_c}\right)/v^{t_c}\right) \times v_s & R > pos^{t_c} \\ pos^{t_c} + \left(t_f - t_c\right) \times v_s & R \leq pos^{t_c} \end{cases}, \tag{15}$$

where $pos^{t_c}$ and $v^{t_c}$ are the position and the velocity of the vehicle, respectively, at the current moment $t_c$; $v_s$ is the velocity limit within the intersection area; and $R$ is the range of the intersection. If $pos_{t_f}$ is beyond the stop line, we consider that the vehicle can pass the intersection in $T_g$. The decision-making system sequentially calculates the position of each vehicle per unit time until $t_e + T_g$ or the number of calculated passable vehicles is equal to $k_i^{T_g}$. Eventually, the sum of the waiting time of these passable vehicles is the eliminable waiting time of the traffic flow.

### 3.4.2. Decision-Making for the Combination Phase and Green Light Duration

As described in Section 3.1, the 8 traffic flows at the intersection can be combined into 12 signal phases ($P_i, \in [1, 12]$) according to the principle of no conflict. Furthermore, we can utilize the equations in the previous section to obtain the eliminable waiting time $ET_i^{T_g}$ of each combination phase.

In addition, if there are few vehicles in a traffic flow, they generally need to wait a longer time for the decision-making system to open their signal phase, which may cause an anxious feeling and a bad experience for the driver. For this point, we adjust the value of $T_d$ to increase the weight gradually with time according to:

$$T_d' = e^{\alpha T_d} - 1, \tag{16}$$

where $T_d'$ is the adjusted waiting time and $\alpha$ is the adjustment factor.

Figure 4 shows the decision-making process of our proposed dynamic phase and period signal control scheme. From $t_e + T_g$ to $t_e + \overline{T_g}$, $t_f$ is sequentially set per unit time to calculate the eliminable waiting time ($ET_{p_i}^{t_f-t_e}$, $t_f \in \left[t_e + T_g, \ t_e + \overline{T_g}\right]$) for all 12 combination phases, and the ratio of the eliminable time of each phase to the total waiting time of all phases ($r^{t_f}$) simultaneously according to:

$$r^{t_e} = \frac{ET_{P_i}^{t_f-t_e}}{WT_s^{t_f}}, \ t_f \in \left[t_e + T_g, \ t_e + \overline{T_g}\right]. \tag{17}$$

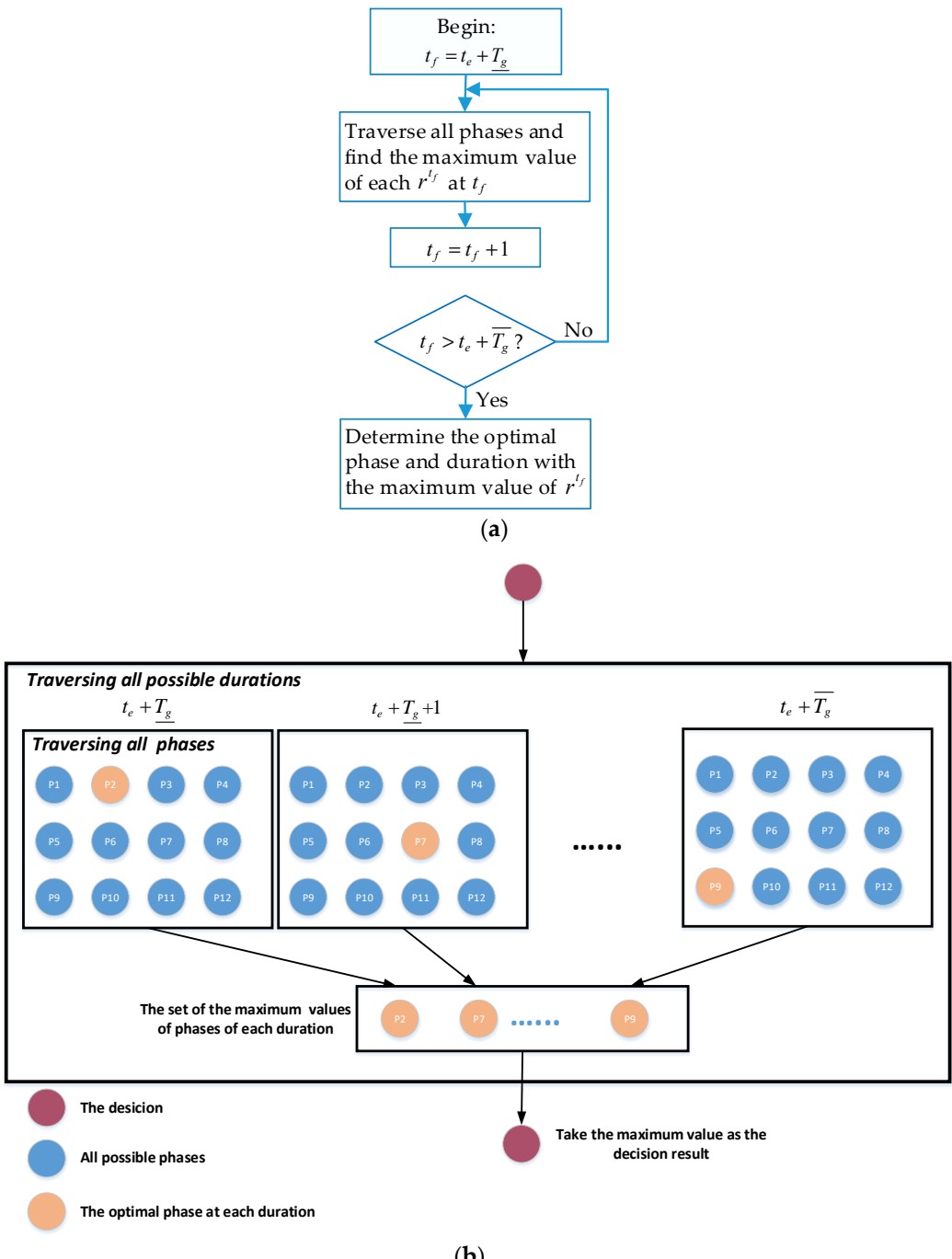

**Figure 4.** Process of decision-making for the combination phase and green light duration: (**a**) decision flowchart and (**b**) decision illustration.

The value of $t_f$ with the maximum value of $r^{t_f}$ is selected as the end time of the next green light, and the corresponding combination phase is the next opening signal phase.

## 4. Simulation and Analysis

### 4.1. Simulation Platform

In this study, Veins (v. 5.0; Heinz Nixdorf Institute, Paderborn, Germany) [34], an open source vehicular network simulation framework was used to evaluate the performance of the above-mentioned intersection signal control mechanism. Based on the open source interfaces of the traffic simulation

software SUMO (v. 1.2.0; German Aerospace Center (DLR), Berlin, Germany) [35] and the network simulation software OMNeT++ (v. 5.5.1; opensim ltd., Las Vegas, USA) [36], the discrete event simulation architecture Veins allowed them to operate synchronously. It could applicably and appropriately simulate the urban traffic operation under a VANET environment.

*4.2. Simulation Environment and Parameter Settings*

First, we constructed a typical intersection environment (as shown in Figure 1), which was formed by four bidirectional six-lane roads with a length of 1 km and an intersection with one left-turn lane and two straight lanes in each entrance direction. The upper bound ($\overline{T_g}$) and lower bound ($T_g$) of green light duration were set to 20 s and 40 s, respectively; the yellow light duration ($T_y$) was set to 3s; and the intersection velocity limit ($v_s$) was 50 km/h. Second, using the OMNeT++ interface, five static network nodes were set up as RSUs at the center of the intersection and the other ends of the four roads. The RSUs could communicate directly with each other and moreover use the 802.11p standard protocol to send ICPs to each adjacent RSU every 2 s in order to collect the information of vehicles on the four roads, where the transmission range of one hop was set to 250 m. Furthermore, the vehicles were generated at the other end of each road toward the intersection, where the lengths of them were set to 5 m uniformly, the safe stopping distance was 2.5 m, and the velocities (km/h) were randomly generated according to a normal distribution N(50,100). According to the intersection environment above and multiple experiments, we adopted 30 veh/min (i.e., 30 vehicles per minute) as the maximum passing capacity of a single lane ($\overline{k_i}$) in the intersection.

In order to evaluate the proposed mechanism thoroughly, in the simulation, three traffic flow environments were provided: (1) The same even traffic flows, which meant all traffic flows had the same average value and their generating time intervals followed the exponential distribution. (2) The same uneven traffic flows, which meant the average traffic flows over a relatively long period were same; however, in order to simulate the actual traffic condition with the signalized intersection, the smooth flows above were separated into several sections according to their different entered flows, i.e., going straight (larger than others), turning left, and turning right. (3) Different traffic flows, for which we selected and set the values of the north-to-south and north-to-east flows to be twice as large as the others to observe the efficiency of the signal control mechanism in the case of differences in traffic flows. The standard value of the traffic flow was set to 150/200/250/300/350 veh/lane/h in five types of simulation tests and each one was run ten times with different random seeds to generate vehicles.

For comparing the performances of our two implementation schemes based on the proposed mechanism assisted by far vehicle information, i.e., the fixed phase and period signal timing improvement scheme (fixed-FA), and the dynamic phase and period signal control scheme (dynamic-FA), the traditional signal timing scheme (traditional timing) that made the decision only considered the intersection vehicles was executed as well. The duration of the simulation was 1800 s. From the simulation results, we adopted the average waiting time of vehicles and the ratio of long-waiting vehicles (waits exceeding 120 s) as evaluation indices to analyze the performance. In addition, in the dynamic-FA, the adjustment factor $\alpha$ for $T'_d$ was set to 0.049, which could reach three times the weight for a waiting time of 120 s.

*4.3. Results and Analysis*

4.3.1. Average Waiting Time of Vehicles

Figure 5 shows the average waiting time of vehicles under different traffic flow environments at the intersection. From the three graphs, it can be clearly seen that the waiting time increased with the increasing traffic volume. Under the environment with the same even traffic flows (Figure 5a), when the value of traffic flows was small, due to the estimation for the arrival time of each far vehicle and then the raising of the precision level in decision-making, the waiting time in the fixed-FA and dynamic-FA was obviously smaller than that in the traditional timing scheme; when the number of vehicles increased,

the results of three schemes tended to be the same (especially around 87 s at 350 veh/lane/h), which was because all arrival traffic flows became equal and saturated, and then the regulating effect of the dynamic signal control dropped gradually and even disappeared regardless of whether the far vehicles were taken into account. Under the environment of same uneven traffic flows (Figure 5b), with the aid of predicting the arriving time of each far vehicle, the signal decision system could make the corresponding adjustment in advance for these uneven traffic flows. Furthermore, compared with the other two fixed-phase signal control schemes (i.e., with four phases), in the dynamic-FA, there were 12 possible combination phases that could provide more available options and enhance the adjustment capability. As a result, the waiting time of vehicles was reduced by averages of 51.4% and 37.8% compared with the traditional timing and fixed-FA schemes, respectively. When the volumes of some individual traffic flows were relatively larger than others (i.e., two times larger), as shown in Figure 5c, the set of values of the waiting time was overall greater than that with the same even traffic flows and the same uneven traffic flows. Moreover, since the signal combination phase and the green light duration were dynamically determined according to the real-time situation, the large traffic flows could be easily favored in the decision-making. Furthermore, with the raising of the accuracy supported by the information of far vehicles, the performance of the dynamic-FA scheme was clearly better than the other two schemes. In addition, Table 2 provides the information of confidence intervals with a 95% confidence level for the calculated average waiting time of the fixed-FA and dynamic-FA schemes when the traffic flow was 250 veh/lane/h.

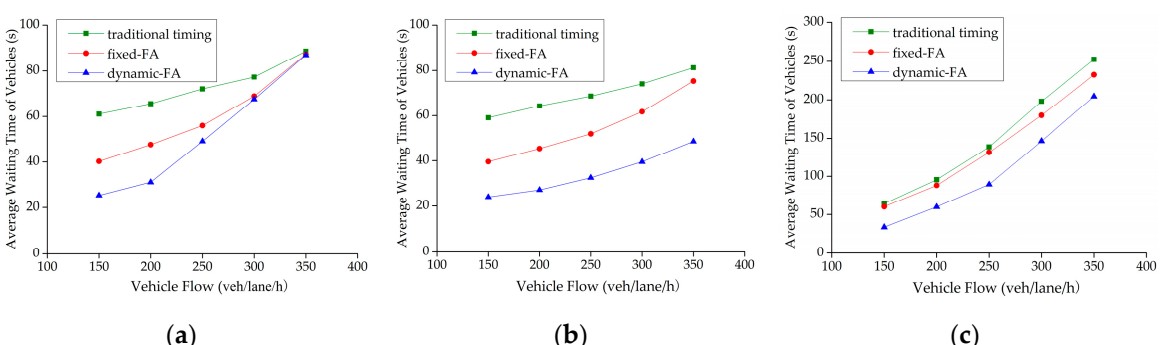

**Figure 5.** Average vehicle waiting time under different traffic environments: (**a**) same even traffic flows, (**b**) same uneven traffic flows, and (**c**) different traffic flows.

**Table 2.** Confidence intervals of the average waiting time.

| Traffic Flow: 250 veh/lane/h | Fixed-FA | | Dynamic-FA | |
|---|---|---|---|---|
| | Average Waiting Time | Confidence Interval (95%) | Average Waiting Time | Confidence Interval (95%) |
| Same even traffic flows | 56.05 s | (54.75, 57.35) | 49.08 s | (45.21, 52.95) |
| Same uneven traffic flows | 51.91 s | (51.19, 52.63) | 32.66 s | (31.32, 34) |
| Different traffic flows | 131.8 s | (125.63, 137.96) | 89.47 s | (83.52, 95.42) |

### 4.3.2. Ratio of Long-Waiting Vehicles

Figure 6 shows the simulation results of the ratio of long-waiting vehicles in three different traffic environments. We can see that the results in the two signal control schemes using far vehicle information was generally smaller than that in the traditional timing scheme, especially the dynamic-FA scheme, which took the long-waiting vehicles into consideration and outperformed the others significantly and enhanced the driver experience substantially. In contrast, based on only the information of the intersection vehicles and without the consideration for long-waiting vehicles, the decision-making in the traditional signal timing scheme had the worst performance, where the average values were even higher than the fixed-FA scheme by 2.6%. In addition, as shown in Figure 6c, some traffic flows had a

large volume. It is remarkable that, when the standard value of the traffic flow was 350 veh/lane/h, the ratios of the long-waiting vehicles in three schemes were close (i.e., 64.9%, 59.4%, and 53.6%); this was because the traffic at the intersection had reached a certain level of saturation and there was no obvious difference among different signal control mechanisms.

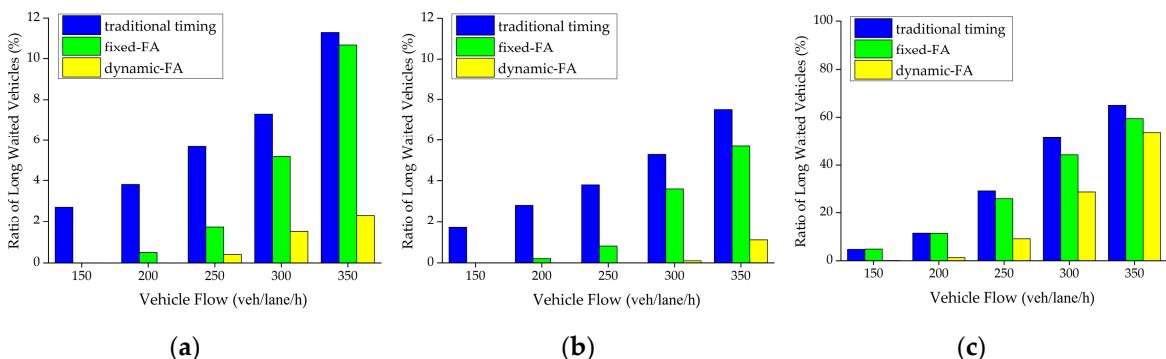

**Figure 6.** Ratio of long-waiting vehicles under different traffic environments: (**a**) same even traffic flows, (**b**) same uneven traffic flows, and (**c**) different traffic flows.

## 5. Conclusions and Future Works

In this paper, we proposed an intersection signal control mechanism that was assisted by obtaining far vehicle information. Through a VANET, the RSU at the intersection collected the real-time information of far vehicles on the road. Incorporating the information of vehicles in the intersection area via image acquisition, the waiting time of each traffic flow at a future moment could be calculated for the signal decision-making. Two concrete schemes were designed, which were a fixed phase and period timing improvement scheme, and a dynamic phase and period control scheme. The latter utilized the eliminable waiting time as the decision reference and took the long-waiting vehicles into consideration. The simulation results show that the two schemes outperformed the traditional signal timing scheme in terms of both the average waiting time of vehicles and the ratio of the long-waiting vehicles. Especially, the dynamic phase and period control scheme showed the best performance.

In the future, we will continue the research on the intersection management under a VANET environment and target the design of traffic priority toward practical needs, such as how to make emergency vehicles pass the intersection smoothly and rapidly.

**Author Contributions:** Conceptualization, Z.C. and M.L.; methodology, Z.C.; validation, Z.C. and Z.D.; investigation, J.Z.; writing—original draft preparation, Z.D. and J.L.; writing—review and editing, Z.C.

**Funding:** The project was supported by the Fundamental Research Funds for the Central Universities under grant 2019MS114 and the Joint Project of the National Nature Science Foundation of China under grant U1636109.

**Conflicts of Interest:** There are no conflicts of interest regarding the publication of this paper.

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
