# Peer review of "An Intersection Signal Control Mechanism Assisted by Vehicular Ad Hoc Networks"

_electronics, doi:10.3390/electronics8121402_

Round 1
Reviewer 1 Report
The reviewer appreciates the effort that the authors put in this manuscript. In general, this idea is interesting, however, more sophisticated methods should have been considered. Some concerns that should be addressed as follows:
- Major issues:
1. The information exchange protocol in VANET presented in Section 2 is not very clear. Specifically, it is stated that "a vehicle selects the closest vehicle in the direction of packet forwarding as the next hop node". However, since the process of getting location information of vehicles was not mentioned, how does a vehicle knows which vehicle is the closest one to it? Moreover, what is the objective of end-to-end connection is Fig. 2? Is it the shortest communication time or information distribution to all vehicles on the road?
2. The method of estimating the locations of vehicles plays an important role in the final results of the proposed intersection signal control mechanism in this manuscript. However, the location estimation presented in Eq. (2) is too simple, thus, the estimated location is not very accurate. The authors should use better location estimation approach to enhance the reliability of the results.
3. Eq. (5) is not correct. The information of traveling time is missing.
4. The operations of the proposed signal control mechanism is quite difficult to follows. It is recommended that the authors should give a figure/table summarizing all possible states and the operations in each state. Moreover, a table summarizing the notations of time related symbol should be provided at the end of Section 1.
- Minor issues:
1. The lower index i of pos_i, e.g., pos_2 and post_5 in line 141 should be defined before using.
2. The authors should remove ":" at the end of sentences.
3. The notation km.h^-1 should be written as km/h.
Author Response
Dear reviewer,
We appreciate your helpful and professional comments. We tried our best to improve the manuscript accordingly. Revised portion are marked in the paper. The point to point responses to the comments are listed as following:
Major issues:
Comment 1: The information exchange protocol in VANET presented in Section 2 is not very clear. Specifically, it is stated that "a vehicle selects the closest vehicle in the direction of packet forwarding as the next hop node". However, since the process of getting location information of vehicles was not mentioned, how does a vehicle knows which vehicle is the closest one to it? Moreover, what is the objective of end-to-end connection is Fig. 2? Is it the shortest communication time or information distribution to all vehicles on the road?
Response: Thanks for your helpful comments. With the respect to the acquisition and exchange of vehicles’ location information in VANETs, we added relevant description (in Section 2) as in most of other VANETs papers.
“…, we assume that each vehicle is equipped with the Global Position System (GPS) and the digital map, and then it can easily acquire the information about its own position, velocity, moving direction, etc. With the aid of the periodical beacon messages (which are mandatory in most of VANETs protocols e.g., 802.11p) exchanged with each other, the information can be also obtained by their neighboring vehicles within the transmission range.”
Fig. 2 illustrates two types of connection between two RSUs at adjacent intersections: 1) the direct connection through wired or cellular networks (line 108 in Section 2); 2) the multi-hop connection through vehicles on the road. The objective of the latter and the delivery of the information collection packet (ICP) on it is to let RSUs obtain the information of all vehicles (including their positions, velocities etc.) on the road in real time, which can support the signal control system to make decision. The former is let RSUs exchange their collected information, especially when the multi-hop connection by the vehicles is disconnected (which means the distance between two successive vehicles is greater than the transmission range). At this point, by the multi-hop ad hoc vehicular network, RSUs cannot collect the information of all the vehicles on the road. However, through the direct connection between them, they can obtain the collected information of each other as much as possible.
Of course, vehicles in the “island” as shown in Fig. 2 (b) are out of contact at current time. Therefore we need to estimate their current locations in Eq. (1) and (2).
Comment 2: The method of estimating the locations of vehicles plays an important role in the final results of the proposed intersection signal control mechanism in this manuscript. However, the location estimation presented in Eq. (2) is too simple, thus, the estimated location is not very accurate. The authors should use better location estimation approach to enhance the reliability of the results.
Response: Sincere thanks for your attention and advice. The locations of the vehicles that can be communicated through multi-hop connection are collected by periodical ICPs in real-time. However, as the vehicles in the “island” as shown in Fig. 2 (b) are out of contact at current time, their locations need to be estimated. According to your professional comment and referring the research in the following paper, we improved the location estimation approach for the “island” vehicles considering the traffic density and the desired velocity in Eq. (1) and (2). Furthermore, with the improved approach, we re-executed the simulation tests which show that the new result data has a better performance compared to the original one. Thanks again for your professional advice and helpful support.
Guo, C.; Li, D.; Zhang, G.; Zhai, M. Real-time path planning in urban area via vanet-assisted traffic information sharing. IEEE Transactions on Vehicular Technology 2018, 67, 5635-5649.
Comment 3: Eq. (5) is not correct. The information of traveling time is missing.
Response: Thanks for your comment. As Eq. (5) is to calculate vehicle’s position at next unit time and the unit time in this paper is set to 1s, we omit the time. After careful thought on your meticulous comment, we added Δt as the unit time and improved the equation.
Comment 4: The operations of the proposed signal control mechanism is quite difficult to follows. It is recommended that the authors should give a figure/table summarizing all possible states and the operations in each state. Moreover, a table summarizing the notations of time related symbol should be provided at the end of Section 1.
Response: Thanks for your reminding. After careful consideration, we added two figures (Fig. 3(b) and Fig. 7(b)) for illustrating the signal phases and the decision-making process. Moreover, in order to enhance the readability of this paper, a table summarizing the presented symbols is provided at the end of Section 1.
Minor issues:
Comment 1: The lower index i of pos_i, e.g., pos_2 and post_5 in line 141 should be defined before using.
Comment 2: The authors should remove ":" at the end of sentences.
Comment 3: The notation km.h^-1 should be written as km/h.
Response: We have rechecked the whole manuscript and corrected above mistakes religiously. Sincere thanks for your careful comments again!
Reviewer 2 Report
The work is interesting, it has a well defined contribution and provides an evaluation.
The concepts are well introduced. The problem formulation is well described and illustrated by figures.
The technical depth of the paper is appropriate for the knowledgeable individual working in the field.
The paper is well structured and well written.
Just some minor points:
- Authors should discuss more recent works.
- The simulator results were validated?
- The selection of values for input simulation parameters should be justified.
- Information about confidence intervals should be included.
- What will be the behaviour of the proposed schemes with different scenarios, node densities, mobility models, etc?
- Some references are incomplete, abbreviated, or not properly formatted.
Author Response
Dear reviewer,
We appreciate your helpful and professional comments. We tried our best to improve the manuscript accordingly. Revised portion are marked in the paper. The point to point responses to the comments are listed as following:
Comment 1: Authors should discuss more recent works.
Response: Thanks for your kind advice. We added and discussed some recent references related to our works. And the proportion of references in recent three years is over 60% in the revised manuscript.
[*] Bi, Y.; Lu, X.; Sun, Z.; Srinivasan, D.; Sun, Z. Optimal Type-2 Fuzzy System for Arterial Traffic Signal Control. IEEE Transactions on Intelligent Transportation Systems 2018, 19, 3009-3027.
[*] Lu, K.; Hu, J.; Huang, J.; Tian, D.; Zhang, C. Optimization model for network progression coordinated control under the signal design mode of split phasing. IET Intelligent Transport Systems 2017, 11, 459-466.
[*] Wang, J.; Jiang, C.; Han, Z.; Ren, Y.; Hanzo, L. Internet of Vehicles: Sensing-Aided Transportation Information Collection and Diffusion. IEEE Transactions on Vehicular Technology 2018, 67, 3813-3825.
[*] Wu, H.; Horng, G. Establishing an Intelligent Transportation System with a Network Security Mechanism in an Internet of Vehicle Environment. IEEE Access 2017, 5, 19239-19247.
[*] Chen, C.; Liu, X.; Qiu, T.; Liu, L; Sangaiah, A. Latency estimation based on traffic density for video streaming in the internet of vehicles. Computer Communications 2017, 111, 176-186.
[*] Khaliq, K. A.; Chughtai, O.; Shahwani, A.; Qayyum, A.; Pannek, J. Road Accidents Detection, Data Collection and Data Analysis Using V2X Communication and Edge/Cloud Computing. Electronics 2019, 8, 896.
Comment 2: The simulator results were validated?
Response: In this paper, aiming at the issue of optimal decision-making for intersection signal control, we design a real-time information collection scheme for the far vehicles by the means of VANETs, and take them into consideration as well as the information of intersection vehicles to determine the intersection traffic condition. Two implementation schemes based on the signal control mechanism are proposed, i.e., fixed phase & period timing improvement scheme and dynamic phase & period control scheme.
In order to evaluate two proposed mechanisms thoroughly, in the simulation software (i.e., SUMO), we set up three different traffic environments (same even traffic flows, same uneven traffic flows and different traffic flows), calculated two performance evaluation indices (average waiting time of vehicles and ratio of long waited vehicles), compared with the traditional signal timing scheme. The index of average waiting time of vehicles can show the throughput capacity of the intersection which is the main objective of the signal control. The index of ratio of long waited vehicles can show the consideration for the individual vehicles which belong to the sparse flow and cannot access to right of way even they have waited for long time. The balance between the whole and the individuals can be supported in our proposed schemes. In addition, the duration of simulation is 1800 s. And for each index, in simulation, we executed 150 times with different random seeds to validate the accuracy and enhance the reliability of the results.
Comment 3: The selection of values for input simulation parameters should be justified.
Response: In this issue, without loss of generality, some common parameters (e.g., traffic volume, vehicle velocity, duration of signal lights, VANETs protocols) are adopted in the simulation. And similar values of these parameters are used in most of related papers. However, we rechecked the whole manuscript and made up the lost description about the value of adjustment factor α (Eq. 16) at the end of section 4.2. Thanks again for your helpful comment.
Comment 4: Information about confidence intervals should be included.
Response: According to your comment, for the primary simulation result (i.e., vehicle waiting time) of our two proposed schemes, we added the information about confidence intervals which can enhance the readability and comprehensibility. Thanks again for your helpful advice.
Comment 5: What will be the behavior of the proposed schemes with different scenarios, node densities, mobility models, etc.?
Response: In the evaluation section, we set up three different traffic environments (same even traffic flows, same uneven traffic flows and different traffic flows) considering different vehicle distributions, different traffic flows and different velocities (i.e., node densities in VANETs). For the mobility model, we only consulted the microscopic vehicle movement models used in the SUMO. As the design of mobility model is not the focus in this research, we did not give a deep research on it. However, according to your professional comment, we will study on this problem in the next step. We sincerely hope you keep your attention on our future work.
Comment 6: Some references are incomplete, abbreviated, or not properly formatted.
Response: Thanks for your careful remainder. In revised manuscript, we have rechecked all references and corrected mistakes consulting the standard in the template.
Round 2
Reviewer 1 Report
After carefully checking the manuscript, the reviewer sees that the authors put a lot of efforts to address most of the previous concerns. The revised manuscript is much better than the previous one.
However, the reviewer is still not satisfied with the modified algorithm which is used to estimate the location of vehicles in the 'island', i.e. using R in the pos_2 + R or pos_5 - R may result in an inaccurate location information compared with the actual one. More importantly, although the authors claimed that the modified algorithm was used, there is no difference in the final results (Figs. 5 and 6) of this revised manuscript and those in previous version. Thus, I suggest that the authors should seriously spend enough time to this matter, then this manuscript may be recommended for publication.
Round 3
Reviewer 1 Report
The reviewer thanks the authors for giving detailed responses and is highly appreciate the efforts that your team put in this manuscript. It is suggested that the algorithm for estimating the positions of island vehicles should be further improved in the future manuscripts.
After considering all aspects, the reviewer is pleased to recommend this manuscript to be published in current form.
Please keep up the hard work!